

# Morphological and molecular barcode analysis of the medicinal tree *Mimusops coriacea* (A.DC.) Miq. collected in Ecuador

Katherine Bustamante[1], Efrén Santos-Ordóñez[2,3], Migdalia Miranda[4], Ricardo Pacheco[2], Yamilet Gutiérrez[5] and Ramón Scull[5]

[1] Facultad de Ciencias Químicas, Ciudadela Universitaria "Salvador Allende," Universidad de Guayaquil, Guayaquil, Ecuador
[2] Centro de Investigaciones Biotecnológicas del Ecuador, ESPOL Polytechnic University, Escuela Superior Politécnica del Litoral, ESPOL, Guayaquil, Ecuador
[3] Facultad de Ciencias de la Vida, ESPOL Polytechnic University, Escuela Superior Politécnica del Litoral, ESPOL, Guayaquil, Ecuador
[4] Facultad de Ciencias Naturales y Matemáticas, ESPOL Polytechnic University, Escuela Superior Politécnica del Litoral, ESPOL, Guayaquil, Ecuador
[5] Instituto de Farmacia y Alimentos, Universidad de La Habana, Ciudad Habana, Cuba

Corresponding author
Efrén Santos-Ordóñez,
gsantos@espol.edu.ec

## ABSTRACT

**Background:** *Mimusops coriacea* (A.DC.) Miq., (Sapotaceae), originated from Africa, were introduced to coastal areas in Ecuador where it is not extensively used as a traditional medicine to treat various human diseases. Different therapeutically uses of the species include: analgesic, antimicrobial, hypoglycemic, inflammation and pain relieve associated with bone and articulation-related diseases. Furthermore, *Mimusops coriacea* could be used as anti-oxidant agent. However, botanical, chemical or molecular barcode information related to this much used species is not available from Ecuador. In this study, morphological characterization was performed from leaves, stem and seeds. Furthermore, genetic characterization was performed using molecular barcodes for *rbc*L, *mat*k, ITS1 and ITS2 using DNA extracted from leaves.
**Methods:** Macro-morphological description was performed on fresh leaves, stem and seeds. For anatomical evaluation, tissues were embedded in paraffin and transversal dissections were done following incubation with sodium hypochlorite and safranin for coloration and fixated later in glycerinated gelatin. DNA extraction was performed using a modified CTAB protocol from leaf tissues, while amplification by PCR was accomplished for the molecular barcodes *rbc*L, *mat*K, ITS1 and ITS2. Sequence analysis was performed using blast in the GenBank. Phylogenetic analysis was performed with accessions queried in the GenBank belonging to the subfamily Sapotoideae.
**Results:** Leaf size was 13.56 ± 1.46 × 7.49 ± 0.65 cm; where is a macro-morphological description of the stem (see Methods). The peel of the seeds is dark brown. Sequence analysis revealed that amplicons were generated using the four barcodes selected. Phylogenetic analysis indicated that the barcodes *rbc*L and *mat*K, were not discriminated between species within the same genus of the subfamily Sapotoideae. On the other hand, the ITS1 and ITS2 were discriminative at the level of genus and species of the Sapotoideae.

## INTRODUCTION

In the genus *Mimusops* (Sapotaceae), 45 species have been described that are distributed in Asia, Africa, Australasia and Oceania. In Ecuador there is no official record of the number of introduced species. Although *Mimusops coriacea* (A. DC.) Miq., has been cultivated widely in the tropics for centuries, it is native only to Madagascar and the Comoro Islands (*Database of tropical plants, 2019*). In Ecuador it has a restricted distribution along the coastal regions.

*Mimusops* spp. are trees reaching a height of up to 25 m, with a dense cope and an irregular short trunk, which exhibit a cracked bark structure. The tree contains simple leaves that are a brilliant green color. Leaves show thick and leathery texture, glabrous, with the central nerve highlighted and 10–20 pairs of lateral nerves. Fruit containing one to several ellipsoid seeds, yellowish brown (*Sánchez, 2011*).

This species is used for various medicinal purposes: the decoction of the stems is considered useful as a tonic and febrifuge; the tender stems are useful in the treatment of urethrorrhea (*Baliga et al., 2011*), cystorrhea, diarrhea and dysentery (*Semenya, Potgieter & Erasmus, 2012*). Traditionally in Ecuador, *Mimusops coriacea* is used as an analgesic and anti-inflammatory (*Erazo, 2010*).

For the genus *Mimusops*, different pharmacological properties have been indicated including antioxidant (*Gillani & Shahwar, 2017*), anti-inflammatory (*Konuku et al., 2017*), antimicrobial activities (*Kiran Kumar et al., 2014*) and hypoglycemic activity (*Saradha et al., 2014*). *Mimusops coriacea* is an important medicinal species in Ecuador; however, little is known about the morphological and anatomical characteristics of leaves, stems and seeds; as well as the molecular barcode. Molecular barcodes will be as a complement for proper species identification. Several molecular barcodes have been used in medicinal plants for these purposes (reviewed by *Techen et al., 2014*); including *rbc*L, *mat*K, ITS1 and ITS2. Although differentiation at the species level is not suitable by using the *rbc*L and *mat*K; the ITS have shown to discriminate at the species level (*Techen et al., 2014*; *Zhang et al., 2016*). Furthermore, barcodes could be used to study patterns of diversifications of the Sapotaceae (*Armstrong et al., 2014*) and for phylogenetic relationships of different genera (*Gautier et al., 2013*). The morphological and molecular barcode characteristics of *Mimusops coriacea* will support subsequent chemical and pharmacological studies, especially for morphological and molecular validation and phylogenetic studies.

## MATERIALS AND METHODS

### Study area

Plant material was collected during May 2018 at the "Botanical Garden," a protected natural vegetative area located in the North zone of "Las Orquídeas" area, next to the Ave. Francisco de Orellana Avenue, in the hills of "Cerro Colorado" of Guayaquil city, Guayas province, Ecuador (coordinates 02°12′13.6800″S 079°53′50.6400″W). The area is located in an altitudinal belt between 50 and 200 m. a. s. l. in a tropical dry forest climate,

with alluvial and sedimentary soils, cumulative rainfall of 1,150 mm/year, with monthly average temperatures of 31.1 °C in winter and 22.6 °C in summer, mean relative humidity of 72% and total evaporation of 1,638.7 mm/year (*Rocero et al., 2010*).

## Morphological analysis

Samples were collected from three adult plants identified by a botanist. Trees approximately 30 m in height, with flowers and fruits were selected via random sampling. One branch containing leaves, fruits and flowers is placed at the GUAY herbarium of Guayaquil University, where the botanists analyzed the samples with taxonomic characters, following proper classification and assignation of a number. Samples from *Mimusops coriacea* were assigned the accession number 13111 in the herbarium.

Morphological description of different organs was performed on fresh and mature leaves ($n = 100$), stems and seeds with a stereoscope (model: Zeizz LUMAR.V12, adapted with an ACXION MRc5 camera). AXION VISION Rel 4.8 (Zeizz, Oberkochen, Germany) software was used in, accordance to the method of *Miranda & Cuéllar (2000)* to analyze leaf ($n = 100$) shape, edge, apex, base, petiole, venation, consistency and color. Size was measured in micrometer. For the stems, the characteristics analyzed includes shape, color, external and internal surfaces and fracture. For fruit characterization, 60 fruits and extracted seeds were analyzed in shape and dimensions, seed coat and endosperm.

For histological analysis, transversals cuts of fresh leaves were performed manually, which were hydrated and clarified with 1% sodium hypochlorite. Tissues were colored with 1% safranin in water, following fixation with glycerinated gelatin according to *Gattuso & Gattuso (1999)*. To analyze anatomical aspects of the leaf epidermis, a longitudinal cut followed with a diaphanization technique was performed. Cleared leaves were obtained with sodium hypochlorite following incubation with 1% safranin in water. Micro-morphological characteristics of cortex were performed to the drug in powder, performing histochemical reactions including: starch determination (Lugol reagent), lignine (1% saphranine in water) and essential oil (5% Sudan III solution in 70% ethanol) (*Gattuso & Gattuso, 1999*). Micromorphology of seeds was performed using dried fragmented material following the procedure described above for leaves and cortex.

## DNA extraction and PCR

Leaves from collected samples from one specimen were ground using liquid nitrogen in the grinder MM400 (Retsch, Haan, Germany) and stored at −80 °C upon DNA extraction. Approximately, 100 mg of leaf was used for DNA extraction using a CTAB protocol with some modifications (*Pacheco Coello et al., 2017*). PCR was performed using the 2× GoTaq® master mix (Cat. # M7123; Promega, Madison, WI, USA) using 0.5 μM of each primer (Table 1). The final volume was 50 μl per reaction. PCR conditions were 95 °C to start denaturation; 35 cycles of: 95 °C for 30 s, 60 °C (for *rbc*L) or 56 °C (for *mat*K, ITS1 and ITS2) for 30 s, 72 °C for 90 s, with a final extension of y 72 °C for 5 min. Five μl of PCR reaction was loaded on a 1.5% gel to check for the presence of amplicons. The remaining 45 μl were purified using the Wizard SV Gel and PCR Clean-Up System

**Table 1 Primers used for amplification of *rbc*L, *mat*K, ITS1 and ITS2.**

| Primer pairs | Sequence | Estimated size (bp) | Locus | Reference |
|---|---|---|---|---|
| rbcLA_F/ | ATGTCACCACAAACAGAGACTAAAGC | 550 | *rbc*L | Costion et al. (2011) |
| rbcLA_R | GTAAAATCAAGTCCACCRCG | | | |
| matK_3F_KIM | CGTACAGTACTTTTGTGTTTACGAG | 850 | *mat*K | Costion et al. (2011) |
| f/matK_1R_KIM R | ACCCAGTCCATCTGGAAATCTTGGTTC | | | |
| ITS 5a F/ | CCTTATCATTTAGAGGAAGGAG | 700 | ITS1 | Chen et al. (2010) |
| ITS 4 R | TCCTCCGCTTATTGATATGC | | | |
| S2F/ | ATGCGATACTTGGTGTGAAT | 400 | ITS2 | Chen et al. (2010) |
| S3R | GACGCTTCTCCAGACTACAAT | | | |

(Cat. # A9282; Promega, Madison, WI, USA) and sequenced commercially (Macrogen, Rockville, MD, USA). At least three technical replicates were sequenced and a consensus was developed.

## Bioinformatics analysis of sequences

Sequences were trimmed from low quality using FinchTV or Chroma's 2.6.5 (*Technelysium, 2018*). Processed sequences were blast (*Zhang et al., 2000*) in the GenBank using the nucleotide database. Sequences from the Subfamily Sapotoideae were selected (GenBank) for phylogenetic analysis using MEGA 7.0.26 (*Kumar, Stecher & Tamura, 2016*) including *Mimusops caffra* (HF542847.1), *Mimusops elengi* (KF686246), *Palaquium amboinense* (HF542854), among others. For each barcode, the recommended model from the MEGA7 was used for the phylogenetic analysis after alignment with MUSCLE. For the phylogenetic analysis, the Maximum Likelihood method was used for each barcode using bootstrap test (100 replicates).

## RESULTS

### Morphological evaluation of the leaves

The leaves were oblong with a coriaceous-waxy texture, containing a short petiole, retuse apex, entire border and an obtuse base. Macroscopic details of the leaves are illustrated in Fig. 1. In respect to the dimensions of the leaves ($n = 100$), the average value observed for the length of the leaves was $13.56 \pm 1.46$ and $7.49 \pm 0.65$ cm for the width.

### Morphological evaluation of the crust

The crust presented a rugose cuticle with an intense gray color, and a slightly brown outer abaxial surface (Fig. 2A) with rough streaks. The internal surface was reddish brown, fibrous and furrowed (Fig. 2B).

### Morphological evaluation of the seeds

In the macro-morphological study, the length and width of the green and ripe fruits ($n = 60$), the seeds ($n = 100$) with the husk and the endosperm of the seeds were considered (Fig. 3). The fruit is rounded and contains one or two seeds. The seeds with a peel are dark brown. The dimensions are presented in Table 2.

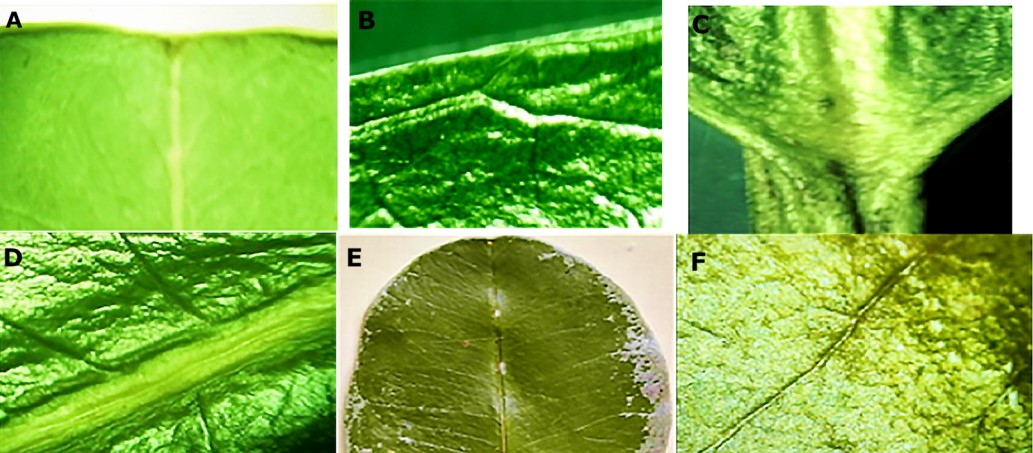

**Figure 1 Macro morphological details of leaf from *M. coriacea*.** (A) Retuse apex, (B) whole edge, (C) obtuse base, (D), (E) and (F) closed rib.

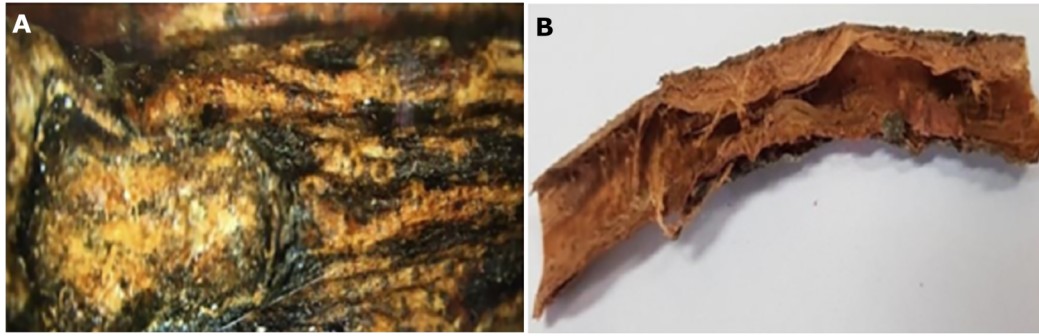

**Figure 2 Macro morphological details of crust from *M. coriacea*.** (A) External surface, (B) internal surface.

## Anatomical evaluation

### *Leaves*

In the leaf anatomy at the level of a cross section of the central nerve (Fig. 4A) the adaxial surface is convex, slightly wavy with the abaxial face concave. An enlarged view of the nerve (Fig. 4B) shows a cuticle of waxy texture that covers the entire leaf, and well visible in the macro-morphological study, followed by the epidermis, which is made up of tabular cells, which gives way to a set of cells that form the spongy parenchyma, given the intercellular spaces which are defined. Possible crystals of calcium oxalate are also observed.

Bordering the central part of the central nerve, there is a cord (Fig. 4C), colored red, corresponding to the endodermis, the structure that surrounds the pericycle. In the middle the conductive tissue formed by the vascular system xylem and phloem is observed (Fig. 4C).

An image of the leaf mesophyll (Fig.4D) shows a somewhat thick cuticle on the abaxial surface, followed by the epidermis, a parenchyma palisade with elongated cells that at times become stratified. In the same way, the entire center of the structure occupied by the

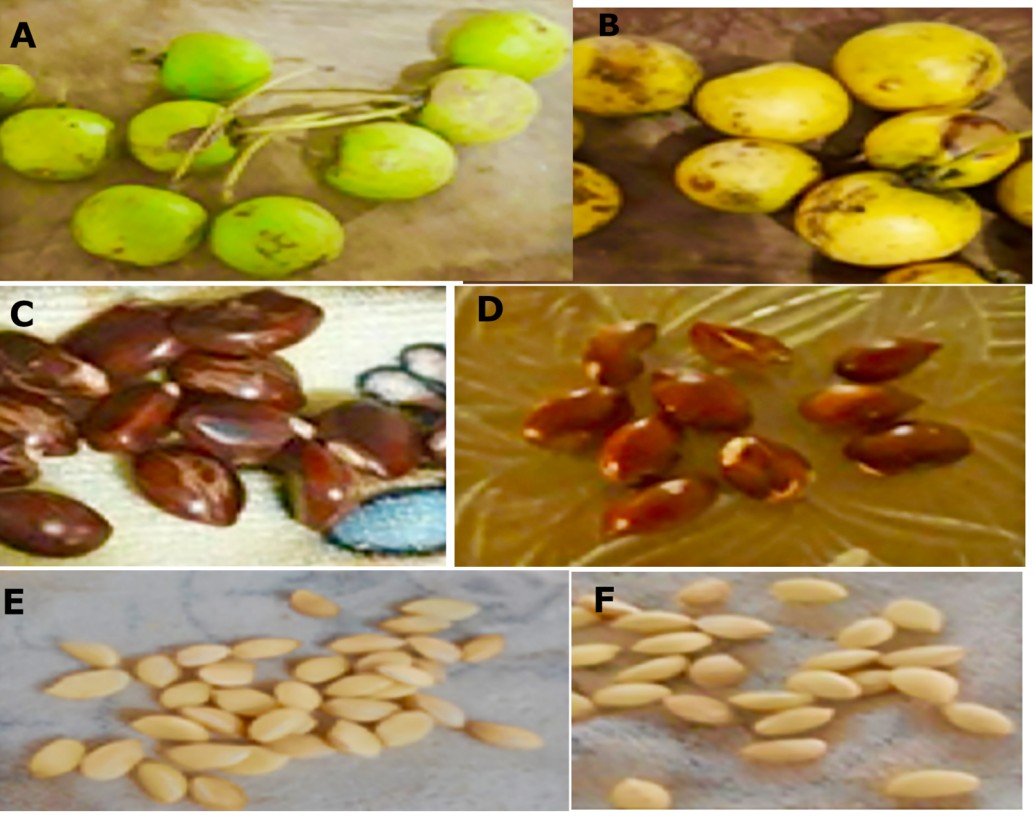

**Figure 3 Macro morphological characters of fruits and seeds from *M. coriacea*.** (A) Green fruit, (B) ripe fruit, (C) seeds green fruits with peel, (D) seeds ripe fruits with peel, (E) endosperm green seeds, (F) endosperm mature seeds.

**Table 2 Dimensions of the fruits and seeds of *M. coriacea*.**

| Type of fruit or seed | Length cm | Width cm |
|---|---|---|
| Green fruit | 2.97 ± 0.18 | 3.14 ± 0.25 |
| Ripe fruit | 2.89 ± 0.2 | 2, 97 ± 0.25 |
| Green seeds | 1.66 ± 0.13 | 1.15 ± 0.21 |
| Ripe seeds | 1.79 ± 0.09 | 1.20 ± 0.09 |

spongy parenchyma is observed, which borders on the upper epidermis that ends with the cuticle, previously mentioned.

The diafanization of a portion of the leaf by the adaxial side showed an epidermis with cells of variable shape and size (Fig. 5A). However, the abaxial epidermis contains a large number of anomocitic type stomata, where the epidermal cells surrounding the pair of occlusive cells are not morphologically different from the rest of the epidermal cells (Fig. 5B). A stain with Sudan III reagent at the level of the epidermis, allowed the visualization of bags with essential oils, which took a reddish coloration (Fig. 5C).

The microscopic analysis of the powder drug showed different fibers and vascular bundles, in this case belonging to the xylematic tissue, classified as scalariform. Figure 5 shows the observed microscopic characteristics.

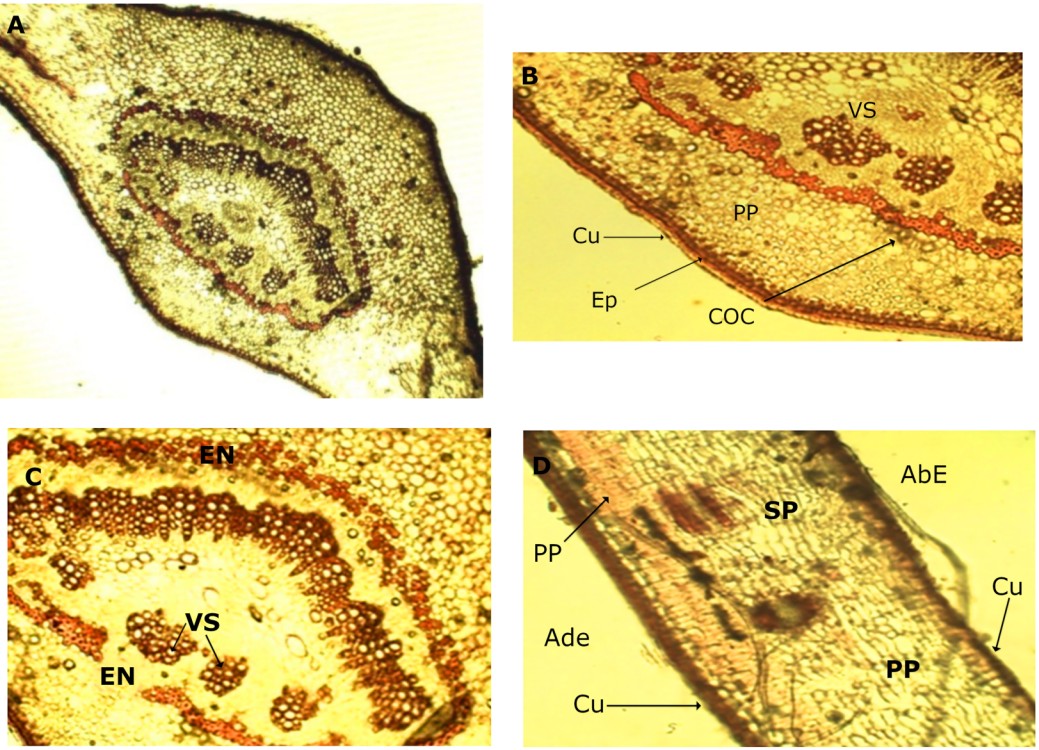

**Figure 4 Microscopic characteristics of leaf from *M. coriacea*.** Transversal section of the central nerve of the leaf (I): (A) central nerve of the leaf, (B) and (C) enlarged view of the central nerve, (D) mesophilic. Cu, cuticle; Ep, epidermis; COC, calcium oxalate crystals; SP, spongy parenchyma; VS, vascular system; En, endodermis; AdE, adaxial epidermis; PP, palisadeparenchyma; AbE, abaxial epidermis.

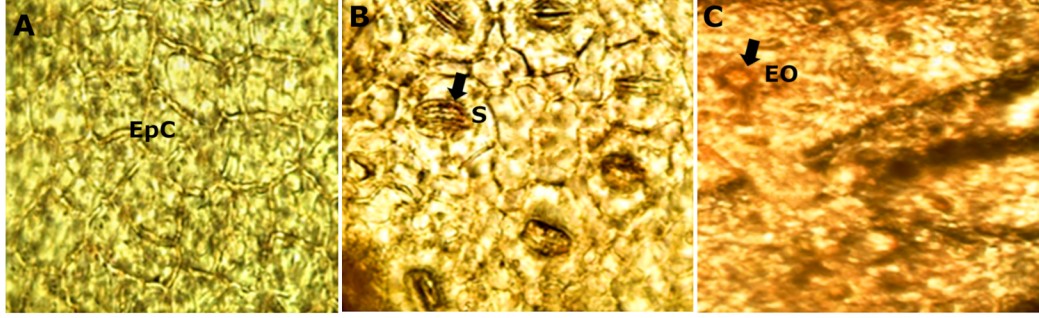

**Figure 5 Microscopic characteristics of leaf from *M. coriacea*.** Diafanized of the leaf (II): (A) adaxial epidermis, (B) and (C) abaxial epidermis. EpC, epidermal cells; S, stomata; EO, essential oils.

## Bark

The micro-morphological analysis of the powder drug showed different fibers and the vascular system, belonging to the xylematic tissue. The xylematic vessels are classified as scalariform (Fig. 6).

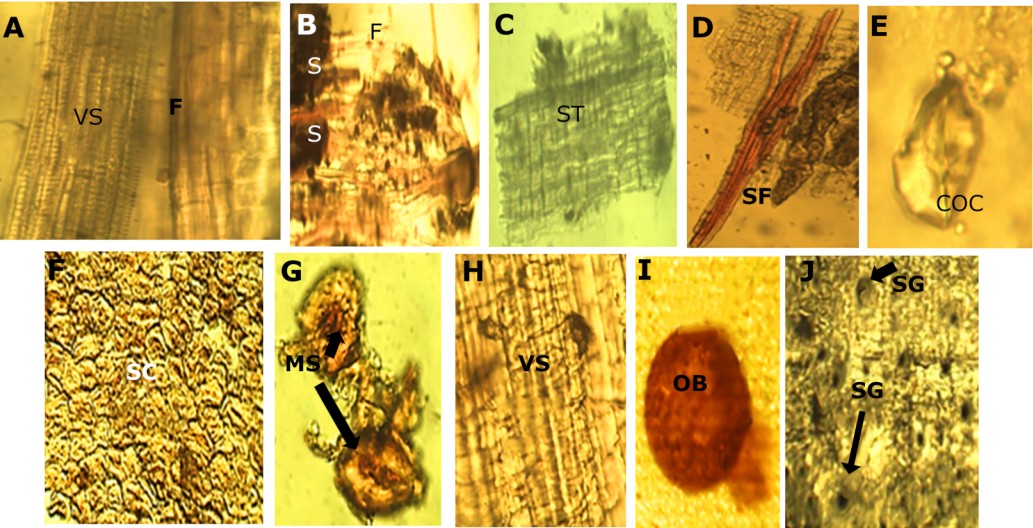

**Figure 6 Powder drug characteristics of *M. coriacea*.** (A) Powder drug from leaf. (B), (C), (D) and (E): powder drug from bark. (F), (G), (H), (I) and (J): powder drug from seed. VS, vascular system; F, fibers; S, starch; ST, suberoustissue; SF, septate fibers; COC, calcium oxalate crystal; SC, sclerides cells; MS, macrosclerides; OB, oilbag; SG, starch granules.

### Seeds

The micro-morphological analysis of the seed powder (Fig. 6), allowed the visualization of a section of the episperm (outer layer of the seed or testa) where the presence of cells of the sclerenchyma tissue corresponding to the supporting tissue is observed. This cell has a well-defined compact arrangement and the walls are slightly thick. The sclerides of the macro-sclerosis type and elements of the conductive tissue was observed. Histochemical reactions on the samples, demonstrated a well-defined red-colored oil pocket that could be observed through the reaction with the Sudan III reagent. Starch granules of ovoid shape and blackish color were observed when using the Lugol reagent.

## Molecular barcode of *Mimusops coriacea*

As a complement analysis for characterization and identification of the *Mimusops coriacea* sample, PCR of the molecular barcodes *rbc*L, *mat*K, ITS1 and ITS2 was performed. Amplicons were detected for all the molecular barcodes (Fig. 7). Sequences were submitted to GenBank (Table 3).

After alignment of the barcode's sequences from GenBank with the *Mimusops coriacea* sample, the best model for phylogenetic analysis are shown (Table 4). The phylogenetic analysis revealed that for the barcodes *rbc*L and *mat*K, most of the *Mimusops* spp. are clustered together with other *genera* (Supplemental Figure). On the other hand, the ITS1 and ITS2 sequences revealed several clades for the different genera including *Mimusops* (Supplemental Figure).

## DISCUSSION

### Morphological evaluation of the leaves

The information referenced in the literature regarding the characteristics of the leaves is limited; thus, comparison with respect to two species of the genus was performed. For

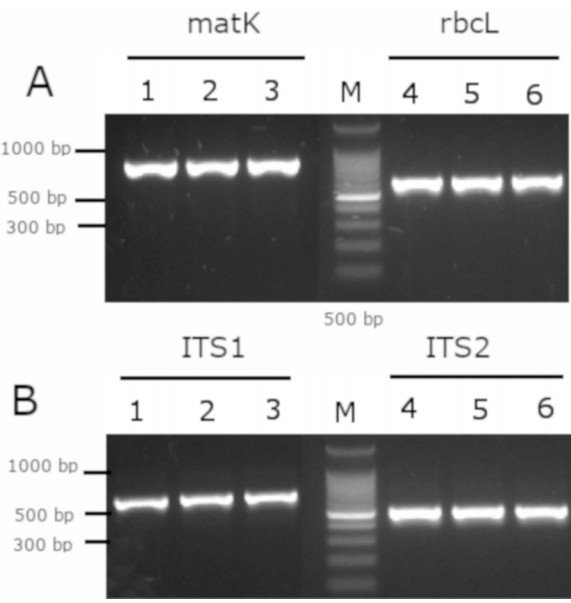

**Figure 7 Gel electrophoresis of amplicons generated for the molecular barcodes with the genomic DNA of *M. coriacea*.** (A) Amplification of rbcL (rbcLA_F/ rbcLA_R), matK (matK_3F_KIM f/matK_1R_KIM R). (B) Amplification of ITS1 (5a_F/ITS 4_R), and ITS2 (S2f/S3R). Numbers (1–3 and 4–6) indicate three technical replicates of DNA. + is the positive control. − is the negative control. M is the 100 bp DNA Ladder (Cat. # G2101; Promega, Madison, WI, USA).

**Table 3 Samples and sequences submitted in the GenBank from the samples of *M. coriacea* barcoded.**

| Barcode | Accession |
|---|---|
| *rbc*L | MK583012 |
| *mat*K | MK583013 |
| ITS1 | MK577640 |
| ITS2 | MK577643 |

**Table 4 Best model to describe the substitution pattern using Mega7.**

| Barcode | Best model |
|---|---|
| *rbc*L | JC |
| *mat*K | T92 |
| ITS1 | T92 + G |
| ITS2 | T92 + G |

Note:
KG, Kimura 2-parameter; +G, Gamma distribution; T92, Tamura 3-parameter; GTR, General time reversible; K2, Kimura 2-parameter; JC, Jukes-Cantor.

*Mimusops elengi* L., *Gami, Pathak & Parabia (2012)* reported that the leaves are elliptical in shape, slightly acuminate at the apex, glabrous with an acute base and petioles 1.3–2.5 cm in length. The dimensions of the leaf range between 6.3–10.0 cm by 3.2–5.0 cm wide, while *Mimusops hexandra* Roxb (without *Manilkara hexandra* Roxb) present oblong leaves,

rounded at the apex, glabrous, dark green in the beam and clear on the abaxial side, with a dimension of 2.5–11 cm long and 1.0–6.0 cm wide (*Chanda, Nagani & Parekh, 2010*). Some species genetically similar to the species under study, present some differences especially in the dimension of the leaf with respect to those study, which are superior.

## Morphological evaluation of the crust

Related to the crust, no referenced information was found.

## Morphological evaluation of the seeds

For the seeds, significant differences were observed between the evaluated parameters of the whole fruits and their seeds at maturity (*Gopalkrishnan & Shimpi, 2011*); for *Mimusops elengi* seed husk was light brown to blackish, with measures of 1.7–1.9 cm long and 1.2–1.5 cm wide, with differs from those obtained for the species studied. The endosperm presented dimensions of 1.42 × 1.0 cm when it came from green fruits and 1.43 × 0.91 cm when it came from ripe fruits, decreasing its thickness in this case.

## Anatomical evaluation

### Leaves

Only for *Mimusops hexandra* Roxb; information about micro-morphological characteristics was found. *Chanda, Nagani & Parekh (2010)* point out similarity regarding the epidermis with rectangular cells, but in their case these were covered with a thin cuticle contrary to that of the species under study that is thick. The stomata of both are anomocitic and more abundant in the abaxial epidermis. Calcium oxalate crystals, spongy tissue with intercellular spaces were also observed. The most marked difference in leaf microscopy is in the form of the central nerve, which in the case of *Mimusops hexandra* is more pronounced toward the abaxial surface than the species under study.

### Crust and seeds

Related to the crust and seeds, no referenced information was found for anatomical characteristics.

## Molecular barcode

Analysis of the molecular barcodes is a complement study for the characterization of the *Mimusops* spp. for medicinal application. Molecular barcode is useful for genotyping organisms, and different *loci* have been proposed characterized land plants (*CBOL Plant Working Group, 2009*). Although, the two proposed *loci* for barcodes are from plastid genome and includes the *rbc*L and *mat*K (*Techen et al., 2014*), other *loci* including ITS1 and ITS2 are widely used for medicinal plants (*Kim et al., 2016*). Furthermore, the ITS2 region is suggested as a barcode for species identification over *rbc*L and *mat*K (*Zhang et al., 2016*). Therefore, the phylogenic analysis for differentiation between genera and species is not practical while using *rbc*L and *mat*K. On the other hand, the ITS1 and ITS2 of the present study were in the same clade as the *Mimusops coriacea* from Madagascar, while the *Mimusops elengi* (accessions KF686246, KF686245, HF542849, KF686245) were in different clades (Supplemental Figure). Furthermore, other molecular barcodes could be

included in future analysis by sampling in different regions in Ecuador; and also by comparing with other results of individual specimens from the family Sapotaceae. Other barcodes may include the plastids *rpl32-trnL, rps16-trnK* and *trnS-trnFM* (*Armstrong et al., 2014*); and *trnH-psbA* spacer, the *trnC-trnD* region (consisting of the *trnC-petN* spacer, the *petN* gene, the *petN-psbM* spacer, the *psbM* gene and the *psbM-trnD* spacer), the *trnC-psbM* region, and the 3′ end of *ndhF* (*Richardson et al., 2014*). However, the ITS is more variable than the plastids barcodes (*Richardson et al., 2014*). Further analysis could be performed to evaluate intraspecific and intraspecific variations of different barcodes to even evaluate at subspecies level.

## CONCLUSIONS

For the first time, the macro and micro-morphological characteristics of the leaves, stems and seeds, of the *Mimusops coriacea* collected in Ecuador were performed. The evaluation of the identity of the species, which is classified taxonomically as *Mimusops* sp., which is a novelty of this work, was confirmed by using molecular barcodes. Most important, the ITS1 and ITS2 indicate more resolution at the species level (*Mimusops coriacea*) than the *rbc*L and *mat*K, confirming published results in medicinal plants. However, further molecular barcode characterization should be performed in *Mimusops* spp. to further validate resolution at the species level as a complement for proper identification using morphological characteristics. Further pharmacognostic analysis will be performed to study medicinal properties of *Mimusops coriacea*.

## ACKNOWLEDGEMENTS

Identification of samples by the GUAY herbarium of the Faculty of Natural Sciences of the Guayaquil University is acknowledged. The study was performed in the framework of the project "*Productos Naturales de interés Agrícola y para la Salud*" (Natural Products of Agricultural and Health Interest) from ESPOL University.

### Funding

The authors received no funding for this work.

### Competing Interests

The authors declare that they have no competing interests.

### Author Contributions

- Katherine Bustamante conceived and designed the experiments, performed the experiments, analyzed the data, prepared figures and/or tables, authored or reviewed drafts of the paper, approved the final draft.
- Efrén Santos-Ordóñez conceived and designed the experiments, analyzed the data, contributed reagents/materials/analysis tools, prepared figures and/or tables, authored or reviewed drafts of the paper, approved the final draft.

- Migdalia Miranda conceived and designed the experiments, analyzed the data, contributed reagents/materials/analysis tools, authored or reviewed drafts of the paper, approved the final draft.
- Ricardo Pacheco performed the experiments, analyzed the data, prepared figures and/or tables, authored or reviewed drafts of the paper, approved the final draft.
- Yamilet Gutiérrez performed the experiments, analyzed the data, prepared figures and/or tables, authored or reviewed drafts of the paper, approved the final draft.
- Ramón Scull performed the experiments, contributed reagents/materials/analysis tools, approved the final draft.

## DNA Deposition

The following information was supplied regarding the deposition of DNA sequences:
Sequences are available at GenBank. rbcl: MK583012, matK: MK583013, ITS1: MK577640, and ITS2: MK577643.

## Data Availability

The DNA sequences are available as a Supplemental File.

## Supplemental Information

Supplemental information for this article can be found online at http://dx.doi.org/10.7717/peerj.7789#supplemental-information.

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
