# Peer review of "Morphological and molecular barcode analysis of the medicinal tree Mimusops coriacea (A.DC.) Miq. collected in Ecuador"

_PeerJ, doi:10.7717/peerj.7789_

## Round 0.1 · original submission · Major Revisions

Please consider the attached reviewers reports and see attached files with suggestions over the document. Please send a rebuttal letter detailing answers to all reviewers concerns.

·

Basic reporting

The paper is severely lacking in the Introduction (not all significant areas introduced), Results (combination of results and discussion) and Discussion (totally absent). Furthermore, numerous unforced errors are found throughout the paper. All these lead to the paper not conforming to expected standard practices for a submitted manuscript. Please refer to the accompanied edited manuscript for additional issues.

Experimental design

The presented experimental design is in accordance with standard practices. However, significant amounts of information is lacking. In addition some information needs clarification. Please refer to the accompanied edited manuscript for additional issues.

Validity of the findings

The findings is presented in accordance with standard practices. Please refer to the accompanied edited manuscript for additional issues.

Additional comments

Please refer to the accompanied edited manuscript for all issues.

·

Basic reporting

• The manuscript is, in general, well written. The use of English language is appropriate and clear.
• The Introduction of the article is not totally coherent with the Abstract. It should be rewritten in order to make it more understandable for the audience. There is information in the Abstract that is missing in the Introduction, for example: Some M. coriacea medical properties and previous studies in M. elengi, thus as the molecular barcodes used in the study.
• The Background could be improved referencing the authors whom studies correspond to general characteristics of Sapotaceae family (lines 69-71) and Mimusops genre (lines 71-77). On the other hand, the Abstract is clear about the importance of the study, but the Introduction must be more explicit about the state of the art of the topic.
• The importance of molecular barcodes, and how it complements the morphological characteristics of the plants for their identification, could be included in the Introduction since it is the basis of the work (lines 83-86).
• At least in the manuscript, the figures are not in a desirable quality.

Experimental design

• As it is redacted in the letter shared by the authors, the plant material was recollected with all the permissions needed and doesn’t affects the ecology or wildlife situation of the studied species.
• The research question and the gap their answer pretend to fill is clearly defined; however, since most of the research is descriptive work, the hypothesis is not well stablished. This point could be improved emphasizing the importance of the molecular barcode used in this study, together with the morphological tissues they measured in the species. Referencing another similar work in related species to M. coriacea.
• Morphological values of plant tissues measured might be presented as Table (and the number of units measured), it would be clearer for the audience and easier to catch the results and characteristics of this species. The methodology to achieve this data seems suitable.
• PCR methodology is well described.
• Bootstrap value of the phylogenetic analysis is not shown.

Validity of the findings

• Morphology measures and PCR information should indicate number of biological and technical replicates.
• Poor quality in Figures 1 to 6 does not provide visually information to confirm the results indicated by the authors.
• Molecular weight marker in gel electrophoresis does not show the ladder values (Figure 7).
• Discussion section is poorly described. A series of suggestions aimed to improve it, are the following:
1. The morphological values obtained from the tissues of M. coriacea should be included in the Discussion section, not in the results as it is written.
2. Authors could contrast the findings in the study with others similar in the field (for example, molecular analysis for the genus or even the species family in order to explain the phylogenetic results) might be done in order to enrich the findings of this study.
3. Complement the achieved results with the ecological context of M. coriacea in the studied region may be interesting to elucidate the natural history of this species, and the differences among its relatives in other regions.
4. Molecular analysis may be the strongest results of the study, it is recommended to enrich this section and make it more relevant for the article.
5. Improve the molecular barcode analysis speculating about other techniques or experiments (for example, recommending alternative molecular markers or comparison between sampling zones).

Additional comments

The knowledge generated by this study is an important base for other works involving common plants in traditional medicine, from a multi-disciplinary view including plant physiology, molecular biology, pharmacology, and even ecology. As a particular opinion, this study has potential for being published, if the authors attend the observations written above. In general, the methodology used in the work is really well applied; however, the manuscript (from the Abstract to the Discussion) is an important area to be improved and to show the real relevance of the work.

---

## Round 0.2 · Minor Revisions

Please check the comments, in particular the quality of the figures, since some of them appear to be out of focus.

·

Basic reporting

Minor grammar corrections needed – refer to accompanied annotated manuscript.
References - Format not consistently applied

Experimental design

Acceptable
Minor grammar corrections needed – refer to accompanied annotated manuscript.

Validity of the findings

Acceptable

Additional comments

Abstract
1. Minor grammar corrections needed – refer to accompanied annotated manuscript.
2. “… and different genus were grouped in one clade of the subfamily Sapotoideae” – unclear statement – rephrase sentence
Introduction
3. Are all of the 1st paragraph from just one source, namely Sánchez (2011)?
4. Minor grammar corrections needed – refer to accompanied annotated manuscript.
5. Focus morphological descriptions only on leaves, stem and seeds –as indicated in the Abstract
6. In Ecuador there is ? species.
7. “Although Mimusops coriacea (ADC – in abstract it is indicated as A.D.C.) Miq,…”
8. In Ecuador it has a restricted distribution along the coastal regions of ….?
9. “This species, as well as others of the genus, is used for different various medicinal purposes…” - This paper focus on LEAVES, STEM and SEEDS (see abstract) – thus ethomedicine should also just focus on these 3 plant structures.
10. Additional issues indicated in the annotated manuscript.

Materials and Methods
11. Acceptable
12. Minor grammar corrections needed – refer to accompanied annotated manuscript.

Results
13. Minor grammar corrections needed – refer to accompanied annotated manuscript.

Discussion
14. “The morphological characteristics of the leaves correspond to that reported by Miranda and Cuéllar (2000) and Gami et al. (2012)” - for which species?
15. Minor grammar corrections needed – refer to accompanied annotated manuscript.
16. “The venation is a closed type, which corresponds to a reticular system (the veins branch and anastomose with each other forming a network that facilitates the diffusion of liquids); which is very common in the dicotyledons. In this case, of the penninervia type, the vascular system is one of the most advanced systems that ensures nutrition to all parts of the leaf (Gami et al., 2012).” – remove - low quality information not suited to high quality international paper.
17. Leaves: The microscopic analysis of the powder drug showed different fibers and vascular
bundles, in this case belonging to the xylematic tissue, classified as scalariform. RESULTS!! Remove.

References
18. Format not consistently applied

Figures
19. Figure 3 – most photos out-of-focus (??)
20. Figure 4 - Cu: cuticle, Ep: epidermis, COC: calcium oxalate crystals – no arrows to indicate location of these structures – see also others.

---

## Round 0.3 · accepted · Accept

The authors have addressed the issues raised during the peer review process. Therefore the manuscript is accepted.